# Expression of Non-T Cell Activation Linker (NTAL) in Jurkat Cells Negatively Regulates TCR Signaling: Potential Role in Rheumatoid Arthritis

**DOI:** 10.3390/ijms24054574

**Published:** 2023-02-26

**Authors:** Isaac Narbona-Sánchez, Alba Pérez-Linaza, Isabel Serrano-García, Inmaculada Vico-Barranco, Luis M. Fernández-Aguilar, José L. Poveda-Díaz, María J. Sánchez del Pino, Fermín Medina-Varo, Mikel M. Arbulo-Echevarria, Enrique Aguado

**Affiliations:** 1Institute for Biomedical Research of Cadiz (INIBICA), 11009 Cadiz, Spain; 2Department of Biomedicine, Biotechnology and Public Health (Immunology), University of Cadiz, 11002 Cadiz, Spain; 3Section of Rheumatology, Puerta del Mar University Hospital, 11009 Cadiz, Spain; 4Puerto Real University Hospital, Puerto Real, 11510 Cadiz, Spain; 5Department of Biomedicine, Biotechnology and Public Health (Biochemistry and Molecular Biology), University of Cadiz, 11002 Cadiz, Spain

**Keywords:** NTAL, LAT2, LAB, rheumatoid arthritis, TCR, signaling

## Abstract

T lymphocytes are key players in adaptive immune responses through the recognition of peptide antigens through the T Cell Receptor (TCR). After TCR engagement, a signaling cascade is activated, leading to T cell activation, proliferation, and differentiation into effector cells. Delicate control of activation signals coupled to the TCR is needed to avoid uncontrolled immune responses involving T cells. It has been previously shown that mice deficient in the expression of the adaptor NTAL (Non-T cell activation linker), a molecule structurally and evolutionarily related to the transmembrane adaptor LAT (Linker for the Activation of T cells), develop an autoimmune syndrome characterized by the presence of autoantibodies and enlarged spleens. In the present work we intended to deepen investigation into the negative regulatory functions of the NTAL adaptor in T cells and its potential relationship with autoimmune disorders. For this purpose, in this work we used Jurkat cells as a T cell model, and we lentivirally transfected them to express the NTAL adaptor in order to analyze the effect on intracellular signals associated with the TCR. In addition, we analyzed the expression of NTAL in primary CD4+ T cells from healthy donors and Rheumatoid Arthritis (RA) patients. Our results showed that NTAL expression in Jurkat cells decreased calcium fluxes and PLC-γ1 activation upon stimulation through the TCR complex. Moreover, we showed that NTAL was also expressed in activated human CD4+ T cells, and that the increase of its expression was reduced in CD4+ T cells from RA patients. Our results, together with previous reports, suggest a relevant role for the NTAL adaptor as a negative regulator of early intracellular TCR signaling, with a potential implication in RA.

## 1. Introduction

T cells recognize antigens in the form of small peptides coupled with Major Histocompatibility Complex (MHC) molecules via a clonally distributed antigen receptor called the T-cell receptor (TCR). Upon TCR binding by a peptide–MHC complex (pMHC), an intracellular signaling cascade is triggered in T cells, leading to their activation and proliferation. Intracellular signals include tyrosine phosphorylation of several proteins, which allows their enzymatic activation or recruitment to different subcellular compartments. One of the first signaling events triggered after TCR engagement is the Lck-mediated phosphorylation of tyrosine residues found on ITAMs present in the cytosolic domains of CD3 [1,2]. Phosphorylated ITAMs recruit the tyrosine kinase ZAP70, leading to its phosphorylation and activation by Lck, and activated ZAP70 phosphorylates the LAT (Linker for Activation of T cells) transmembrane adaptor [3,4]. Tyrosine phosphorylation of LAT generates docking sites for cytosolic adaptors such as Grb2, Gads, and SLP-76, or effectors such as PLC-γ1 or Vav. However, despite the central role of LAT in the transduction of activator signals coming from the TCR complex, analysis of several knock-in mouse strains showed that the LAT adaptor is also capable of negatively regulating T-cell homeostasis [5,6,7,8].

LAT expression is restricted to thymocytes, peripheral T lymphocytes, Natural Killer (NK) cells, mast cells, and platelets, but is not expressed in B cells [9]. NTAL (non-T cell activation linker) is a LAT-like adaptor protein that is expressed in B, NK, and mast cells, which is rapidly phosphorylated after BCR or FcR engagement [10,11]. While NTAL lacks a binding site for PLC-γ1, both adapters are capable of generating binding sites for Gads and Grb2 [10,11], and the three NTAL C-terminal tyrosines are essential for signal transduction [12]. Although LAT knockout results in a block of thymic development and the absence of mature peripheral T cells [13], NTAL knockout mice show no difference in the development or phenotype of the B compartment as compared to normal mice [14]. Interestingly, NTAL knockout eventually developed an autoimmune syndrome characterized by splenomegaly and the presence of antinuclear antibodies [15]. The B cells present in these mice were normal, but T cells showed a hyperactivated phenotype producing high levels of cytokines such as IL-2, IL-10, and INF-γ, demonstrating for the first time that NTAL can negatively regulate T cell activation. This paradoxical phenotype could be explained by the fact that, although NTAL is not expressed in resting T lymphocytes, it is indeed expressed in activated T cells [15]. This regulatory role of NTAL in T cells is consistent with the fact that although an NTAL transgene expressed in a LAT-deficient strain is able to restore T cell development, these mice develop severe organomegaly with hyperactivated T cells, producing large amounts of TH2 cytokines [16]. In addition, NTAL seems to play a negative regulatory function in B cells, since NTAL deficient B cells exhibit slightly increased Ca^2+^ mobilization and proliferation after BCR crosslinking [11,14]. Thus, despite its similarity to LAT, the NTAL adaptor does not exert a function in B cells that is similar to the role played by LAT in T cells [14,16].

Rheumatoid arthritis (RA) is a systemic autoimmune disease characterized by persistent synovitis, inflammation, and autoantibodies against rheumatoid factor and citrullinated peptide, affecting up to 1% of the world’s population [17]. Although the etiology of RA remains unresolved, a genetic component of susceptibility to RA has been firmly established [18]. There are more than 100 confirmed polymorphisms showing a strong association with RA, mainly with MHC class-II alleles, thus reinforcing the central role of adaptive immunity, particularly effector T cells. Among the more than 100 genes implicated in RA predisposition, many of them are involved in T-cell selection, maturation, and function [19,20]. Tumor necrosis factor (TNF) plays a central role in the pathogenesis of RA, and TNF inhibitors effectively suppress inflammatory activity in RA in a percentage of patients [21]. However, the considerable inter-patient variability in response to the current treatments is a challenge, and more biomarkers are needed for patient classification. It has been recently reported that p-p38, IkBa, p-cJun, p-NFkB, and CD86 may constitute markers for differentiation between RA patients and healthy donors [22].

Although the intracellular signals responsible for the activation of T lymphocytes following antigen recognition are well understood, we still lack a thorough understanding of the molecular mechanisms by which tolerance mechanisms are broken down in genetically susceptible individuals. In this context, it can be proposed that autoimmune diseases would be triggered either by an increase in the percentage of autoreactive T lymphocytes, or by defects in any of the peripheral tolerance mechanisms, or by a combination of both. Any of these scenarios would be the result of alterations in the intracellular signaling cascade associated with the TCR, so a detailed analysis of these events could be useful to elucidate the molecular mechanisms underlying RA, and consequently facilitate the discovery of prognostic and/or diagnostic markers. In this report we confirmed that NTAL expression in Jurkat cells negatively affects TCR-mediated signals and that the NTAL adaptor is expressed in human CD4 T cells after CD3/CD28 stimulation. Moreover, comparative analysis showed that the NTAL expression increase in CD4 T cells from RA patients was reduced as compared with healthy controls, and that CD3 stimulation of CD4 T cell blasts from RA patients generated a greater phosphorylation of Erk when compared with CD4 T cells from healthy controls, suggesting that decreased NTAL expression could affect the disease progression. Altogether, our data support a role for NTAL as a negative regulator of early intracellular TCR signaling, which could have relevance in immune-based pathologies.

## 2. Results

### 2.1. Impact of NTAL Expression in Jurkat Cells for ZAP70 Activation

To analyze the potential role of NTAL expression in T cell function, we generated Jurkat cells expressing this adaptor protein by means of a lentiviral system [23]. This system allows the generation of polyclonal cell populations, avoiding the undesired effects of analyzing individual clones. In all cases, since our vector had a GFP reporter behind an IRES sequence [23], the vector expression levels were higher than 80%. As can be seen in Figure 1A, Jurkat cells do not express the NTAL adaptor, and after a lentiviral transfection NTAL adaptor was detected by Western blot. Following TCR engagement, the protein tyrosine kinase ZAP70 is rapidly phosphorylated on several tyrosine residues. Phosphorylation of tyrosine 319, located in the interdomain B of ZAP70, is needed for the transduction of intracellular signals coming from the TCR/CD3 complex [24]. Thus, Jurkat cells expressing NTAL were stimulated at 37 °C with anti-CD3 monoclonal antibody, and whole-cell lysates were obtained after 0, 3, 10, and 30 min of stimulation. As can be seen in Figure 1B, anti-CD3 stimulation induced a rapid increase in the phosphorylation level of Tyr319 of ZAP70 in both Jurkat cells and Jurkat cells expressing NTAL (Jurkat-NTAL). No clear differences in the phosphorylation level of Tyr319 in ZAP70 were observed between Jurkat and Jurkat-NTAL cells. Densitometric quantification of four independent experiments failed to show differences in the activation of ZAP70 (Figure 1C).

### 2.2. NTAL Expression Modulates Calcium Signaling in Jurkat Cells

It has been described that T cells from NTAL knockout mice show increased calcium responses after TCR/CD3-mediated stimulation [15]. Therefore, it was of interest to verify whether the expression of NTAL in Jurkat cells decreases calcium influx generation. To do so, Jurkat and Jurkat-NTAL cells were labeled with the intracellular calcium indicator Indo-1 and then stimulated with anti-CD3 mAb. Before the stimulation, cells were kept at 37 °C for one minute, and then stimulated with anti-CD3 mAb (OKT3, 1 μg/mL). The increase in the ratio of fluorescence emitted at 405 and 485 nm (F405/F485) was collected alternately and was indicative of the rise in intracellular calcium concentration. As can be seen in Figure 2A, the calcium concentration increased after CD3-mediated stimulation in both Jurkat and Jurkat-NTAL cells. The intensity of such increase was similar for both types of cells, but after reaching a peak about two minutes after anti-CD3 stimulation (about 200 s from the start of the experiment), the calcium concentration began to decline slowly, and this decline was more pronounced in Jurkat-NTAL compared with Jurkat cells (Figure 2A). The difference in calcium concentration between Jurkat and Jurkat-NTAL cells reached statistical significance after approximately 5 min (300 s) from the start of the experiment.

We next sought to verify whether the observed calcium reduction was caused by a deficient PLC-γ1 activation. Thus, Jurkat and Jurkat-NTAL cells were stimulated with the OKT3 anti-CD3 mAb and phosphorylation of tyrosine 783 in PLC-γ1 was analyzed by Western blot in the corresponding cell lysates. As can be seen in Figure 2B, CD3 stimulation induced phosphorylation of PLC-γ1-Tyr783, which is indicative of its enzymatic activation [25]. Both Jurkat and Jurkat-NTAL cells showed upregulation of PLC-γ1 phosphorylation at all stimulation time points analyzed (Figure 2B). Interestingly, PLC-γ1 phosphorylation in Jurkat-NTAL cells was reduced compared with Jurkat cells. Densitometric quantification of four independent experiments show that the relative phosphorylation of Tyr783 in PLC-γ1 was statistically higher in Jurkat cells compared with Jurkat-NTAL cells after 10 min of CD3 stimulation. Therefore, these results support the view of NTAL as a negative regulator of intracellular calcium-dependent signaling pathways associated with the TCR/CD3 complex.

### 2.3. Effect of NTAL Expression in Jurkat Cells on the MAP Kinase Signaling Pathway

It has been shown that TCR stimulation leads to Erk activation through two different mechanisms, and both of them need phosphorylation of the LAT membrane adaptor [26]. One of these mechanisms requires the activation of the PLC-γ1-RasGRP1 pathway, while the other depends on the binding of Grb2-SOS to phosphorylated LAT. Given the negative effect of NTAL expression on PLC-γ1 activation, it was of interest to test what happened in the activation of the MAP kinase pathway after activation through the TCR/CD3 complex in Jurkat cells expressing NTAL. Accordingly, Erk phosphorylation at residues Thr202/Tyr204, which is indicative of its enzymatic activation, was analyzed in lysates of Jurkat cells stimulated for different times. As observed in Figure 3A, anti-CD3 stimulation induced rapid Erk phosphorylation, which was clear at 3 min of stimulation, and was still appreciable at 30 min. However, after densitometric quantification of four independent experiments, we were not able to appreciate quantitative differences in the relative Erk phosphorylation levels in Jurkat and Jurkat-NTAL cells (Figure 3A, right graphics).

This result was rather unexpected, given the negative effect that NTAL expression had on PLC-γ1 activation. In order to further test the effect of NTAL expression on the MAP-kinase pathway, we analyzed the activation of MEK, i.e., the kinase responsible for the phosphorylation and activation of Erk [27]. Therefore, we used a specific antibody to phospho-MEK-1/2 to investigate MEK activation. CD3 stimulation of Jurkat cells rapidly induced phosphorylation of MEK, and the same was observed for Jurkat-NTAL cells (Figure 3B). Again, densitometric quantification of four independent experiments did not show any difference between both types of cells (Figure 3B, right panel). Therefore, from these results we can conclude that NTAL expression in Jurkat cells does not modify the activation of the MAP-kinase pathway.

### 2.4. NTAL Expression in Human Peripheral T Cells

It has been previously shown that mouse peripheral T lymphocytes can express NTAL adaptor after activation [15]. Given that NTAL knockout mice develop an autoimmune syndrome with hyperactivated T cells, producing higher levels of cytokines than T cells from wild type mice, it was of interest for us to study if T cells from healthy donors and Rheumatoid Arthritis (RA) patients express this transmembrane adaptor. Thus, we obtained peripheral blood from healthy donors and rheumatoid arthritis patients, and CD4+ T lymphocytes were purified and cultured for 5 days in the presence of anti-CD3/CD28 beads and recombinant interleukin-2 (IL-2). The percentage of CD3+CD4+ T cells from both RA patients and healthy donors was analyzed by flow cytometry, and the average was higher than 85%. Cell lysates were obtained from resting and activated cells, and NTAL expression was determined by Western blot with a specific mAb. Membranes were stripped and subsequently blotted with anti-β-actin mAb in order to determine total protein load and allow comparative analysis. As shown in Figure 4A, samples corresponding to resting cells from both healthy controls (C) and RA patients (P) showed very low, barely detectable levels of NTAL expression, with one exception. The activation of T cells with anti-CD3/CD28 beads plus IL-2 led to an increase in the number of cellular proteins, as demonstrated by β-actin Western blots (Figure 4A, bottom panels). Interestingly, the activation of T cells induced the expression of the NTAL protein (Figure 4A, top panels), demonstrating that the NTAL adaptor is also expressed in activated CD4+ T cells. We performed densitometric quantification of bands and calculated the relative expression of NTAL with regard to β-actin expression for resting and activated CD4+ T cells for 7 healthy controls and 16 RA patients. Every experiment with CD4+ T cells was always performed at the same time with one healthy control, and for quantifications a value of 1.0 was assigned to resting cells from healthy controls. As it can be seen in Figure 4B, resting CD4+ T cells from RA patients showed slightly enhanced relative expression of NTAL compared with CD4+ T cells from healthy controls, although there was not statistical significance. However, the relative expression of NTAL in activated CD4+ T cells was higher in healthy controls compared with RA patients (Figure 4B), although, again, the difference did not reach statistical significance (*p* = 0.1030). Given that the NTAL expression levels were slightly higher in resting cells from RA patients, we decided to calculate the relative increase in the expression of this membrane adaptor in CD4+ T cells from patients and healthy controls. As can be seen in Figure 4C, the increase was greater in healthy controls than in patients, and in this case there was a statistically significant difference (*p* < 0.05). Therefore, the induction of NTAL expression, which appears to act as a brake on T-cell activation, was decreased in rheumatoid arthritis patients relative to healthy controls, which could explain part of the pathology of these patients.

### 2.5. Increased Erk Activation in CD4+ T Cells from RA Patients

Data obtained in mice demonstrated that TCR-mediated signaling was enhanced in T cells from NTAL knockout mice [15]. Interestingly, the same group showed that an NTAL transgene under the control of the human CD2 promoter in mouse T cells produced a significant reduction of TCR-signaling. Our data show that the induction of NTAL expression in CD4+ T cells from RA patients was reduced compared with healthy controls, and this could mean that they would be activated more intensely. Therefore, we purified CD4+ T cells from RA patients and healthy controls and cultured them in the presence of anti-CD3/CD28 and IL-2 for 5 days in order to induce NTAL expression, and then analyzed Erk activation after anti-CD3 stimulation. Figure 5A shows three independent experiments in which CD4+ T blasts were stimulated for 0, 5, and 10 min with anti-CD3 mAb, which induced phosphorylation of Erk. Membranes were stripped and reblotted with anti-Erk mAb to show total protein load for every sample and allow densitometric quantification. The images show that, although variations in Erk phosphorylation were observed, a trend was evident wherein CD4+ T cells from RA patients exhibited a higher induction of Erk phosphorylation (Figure 5A). Five independent experiments were performed, always including a healthy control with samples from RA patients. Quantification of Erk phosphorylation for five healthy controls and nine RA patients did show increased Erk activation in RA patients compared with healthy controls, at both 5 and 10 min, with a statistically significant difference at 5 min of anti-CD3 stimulation. Consequently, this result is concordant with the lower induction of NTAL expression in CD4+ T cells in RA patients, which could be one of the causes of T cell hyperactivation in this pathology.

## 3. Discussion

RA is a chronic inflammatory disease affecting the synovium, leading to joint damage and bone destruction, and causes severe disability and increases mortality [17]. Prevalence studies of RA show a substantial variation of the disease occurrence among different populations, with a prevalence of 0.5–1.1% in Northern European and North American areas, and a relatively lower prevalence of in 0.1–0.5% in developing countries [17]. There is a consensus that RA is a multifactorial disease, resulting from the interaction of genetic and environmental factors. Although the etiology of RA remains unresolved, a genetic component of susceptibility to RA has been firmly established [18]. More than 100 polymorphisms associated with RA have been confirmed, mainly with MHC class-II alleles, pointing to the involvement of adaptive immunity mechanisms in the origin and development of this disorder [19,20]. In this context, it has been suggested that infectious agents could activate adaptive immune responses triggering the development of the disease in genetically susceptible individuals [19].

Therefore, balanced and carefully controlled immune responses are essential to avoid adverse reactions that could lead to autoimmune pathological processes. T cells are central players in adaptive immune responses, and these cells are specifically activated upon antigen recognition through the TCR. After TCR engagement, an intracellular signaling cascade is triggered, leading to the activation and proliferation of naive T cells, as well as their differentiation into effector/memory cells. Regulation of such signals is crucial to prevent exaggerated responses, which might be harmful for the organism. TCR-activating intracellular signals are well understood, but many spatiotemporal and regulatory aspects remain to be discovered [1,2]. In contrast to LAT with respect to T cells, NTAL knockout does not impair the development of B cells, NK cells, or mast cells [14,28]. Remarkably, mast cells from NTAL-deficient mice showed increased degranulation, calcium flux, and cytokine production [29], indicating that, like LAT, NTAL also possesses intrinsic functions to control cell activation. Additionally, aged NTAL knockout mice develop an autoimmune syndrome, with enlarged spleens and production of autoantibodies, which is due to the absence of NTAL expression on activated T cells [15]. This was the first time that expression of this transmembrane adaptor was demonstrated in activated mouse T cells, raising the possibility that NTAL expression could control excessive T cell activation.

We demonstrated that human CD4+ T lymphocytes from healthy donors express the NTAL adaptor after stimulation with anti-CD3/CD28 microbeads plus IL-2. Therefore, this seems to be a general mechanism for the control of immune responses mediated by T lymphocytes. Concordantly, NTAL expression in Jurkat cells negatively modulates calcium influxes and PLC-γ1 activation after CD3 stimulation. In contrast, NTAL expression in Jurkat cells does not modify ZAP70 activation. This is an expected result, since LAT and NTAL adaptors are known substrates of ZAP70 and Syk tyrosine kinases, respectively [1,30]. Moreover, ZAP70 is the only kinase phosphorylating NTAL in T cells, since Syk is not expressed in this population. Therefore, it was not foreseeable that an effector such as ZAP70 upstream of NTAL would be affected in its enzymatic activity. Consequently, the negative role exerted by NTAL expression in PLC-γ1 activation and calcium influx generation may be due to competition with LAT for localization in lipid rafts, as previously suggested by others [15]. It is possible that ZAP70 substrates (LAT and NTAL) compete for space in membrane areas, but one of them (NTAL) lacks a PLC-γ1-binding site, causing its activation to be reduced.

However, as opposed to calcium fluxes and PLC-γ1 activation, Erk and MEK phosphorylation are not affected by the expression of NTAL in Jurkat cells. The latter is a conflicting result with that observed by Zhu et al. in NTAL-deficient mice and transgenic mice expressing NTAL in T cells [15], in which the negative effect of NTAL expression on Erk phosphorylation was demonstrated. One possible explanation for this discrepancy may lie in the difference between the Jurkat cell line and primary T cells. In fact, even the involvement of the LAT adaptor with Erk activation is controversial. We previously showed that Erk phosphorylation is similar in the absence of LAT in J.CaM2 cells (a Jurkat derivative deficient in LAT expression) cells, and the reintroduction of LAT had no substantial effect on Erk phosphorylation [31]. On the other hand, using a conditional knockout mouse strain with normal thymic development but able to generate CD4+ T lymphocytes deprived of LAT [8], Malissen and coworkers showed that CD3-mediated stimulation of CD4+ T lymphocytes did not induce Erk phosphorylation. Besides, Samelson and coworkers demonstrated a LAT-independent pathway by which Erk can be activated after CD3 stimulation [32]. Therefore, given the functional complexity of LAT and NTAL adaptors, the different effects observed in Jurkat and primary T cells are not surprising. More work is needed to shed definitive light on this point.

Given the potential role of NTAL as a negative regulator of T cell activation, we sought to verify whether activated CD4+ T cells from RA patients expressed this transmembrane adaptor or not. We demonstrated for the first time that human T cells are able to express NTAL after CD3/CD28-mediated activation. Our data obtained in Jurkat cells, together with the phenotype of NTAL knockout and NTAL-Tg mice, suggest that in RA patients, in whom T-cell hyperactivation is one of its typical hallmarks [19,20], there would be a deficit of NTAL expression. We found that activated CD4+ T cells from RA patients showed a reduced increase in the expression of NTAL as compared with healthy controls. We are aware that of our work involves a small number of patients. Another limitation is the lack of study on CD8+ T cells. Future work should address, with a larger number of patients, the combined analysis of both populations. However, our data suggest that NTAL expression may act as a brake on excessive T-cell activation. Very little is known about the regulation of NTAL expression at the transcriptional level. It has been shown that the leukaemia-specific fusion oncoprotein RUNX1/RUNX1T1 represses the expression of NTAL [33]. Indeed, knockdown of RUNX1/RUNX1T1 with small interfering RNA (siRNA) decreased NTAL expression in Kasumi-1 cells, a myeloblast cell line widely used as a model for the study of myeloid leukemias [34]. Interestingly, these authors showed that repression of NTAL expression is readily reversed by the use of entinostat and mocetinostat, two histone deacetylase (HDAC) class I-specific inhibitors. If the reduced expression of NTAL in activated T cells from RA patients is confirmed as a key factor in the development and/or progression of this disease, the use of these drugs as a potential therapy for its control would not be ruled out.

Consistent with a reduced increase in NTAL expression in activated CD4+ T cells from RA patients, they show increased activation of Erk phosphorylation. Although we did not observe any modification in Erk or MEK phosphorylation in Jurkat cells expressing NTAL, the Jurkat model is perhaps not suitable for the analysis of the MAP-kinase signaling pathway. Our data obtained in activated CD4+ T cells are consistent with previous reports showing increased excessive activation of the Ras/MEK/ERK pathway in T lymphocytes from RA patients as compared with T cells from healthy donors [35]. In line with that, pharmacological inhibition of Ras GTPases significantly reduced the disease in the rat adjuvant-induced arthritis model (AIA) [36]. The association of RA with a non-synonymous SNP (R620W) missense mutation in the PTPN22 phosphatase has been shown [37]. PTPN22 is a tyrosine phosphatase involved in the regulation of TCR signaling, and this mutation increases its phosphatase activity, lowering the TCR threshold. Here, we presented data pointing to the decreased induction of NTAL expression on CD4 T cells in RA patients. This could be an additive factor for the development and progression of the disease. If confirmed, new therapeutic strategies aimed at increasing the expression of this membrane adaptor on T cells could be explored.

## 4. Materials and Methods

### 4.1. Cell Culture, Cloning and Lentiviral Transfection

Jurkat cells were grown in complete RPMI 1640 medium supplemented with 10% FCS (both from Lonza, Basel, Switzerland), and 2 mM L-glutamine at 37 °C in a humidified atmosphere containing 10% CO_2_.

NTAL cDNA cloning was performed as previously described [23]. The coding sequence in the plasmid was verified by sequencing, and then subcloned in the SIN lentiviral transfer plasmid pHR’SINcPPT-Blast through site-specific recombination (Gateway LR Clonase, Invitrogen, Waltham, MA, USA). Lentiviral supernatants were generated and used to induce NTAL expression in polyclonal cell populations. Blasticidin selection (20 μg/mL) was applied to transduced cells after 72 h of culture, and the expression of GFP was analyzed as a reporter of transfection using FACS analysis (CytoFLEX, Beckman Coulter, Brea, CA, USA). In all cases, the percentage of GFP-positive cells was higher than 80%.

### 4.2. Antibodies and Reagents

Anti-LAT, anti-PLC-γ1, and anti-Erk antibodies were from Santa Cruz Biotechnology (Heidelberg, Germany). The anti-NTAL NAP-07 monoclonal antibody was from EXBIO (Prague, Czech Republic). Antibodies binding phospho-Erk, phospho-PLC-γ1-Tyr783, ZAP70, phospho-ZAP70-Tyr319, phospho-MEK-Ser221, and anti-MEK were from Cell Signaling Technology. Anti-β-actin, anti-CD3 (OKT3), and anti-CD4 monoclonal antibodies were provided by Biolegend (San Diego, CA, USA).

### 4.3. Purification and Culture of CD4+ T-Cells

Ethical approval for this study was obtained from the Comité Coordinador de Ética de la Investigación Biomédica de Andalucía (Spain) (reference numbers 01/2018 and 120.21), and informed consent was obtained from all subjects enrolled. Patients’ and healthy controls’ blood samples were collected in the Rheumatology area of the Hospital Puerta del Mar (Cádiz, Spain). Patients had confirmed written diagnosis of Rheumatoid Arthritis according to ACR/EULAR 2010 criteria. All patients and healthy controls, males and females, were between 41 and 72 years old. RA patients had a mild-to-moderate inflammatory burden, and their treatment consisted of conventional disease-modifying antirheumatic drugs (DMARD). CD4+ T cells were purified from peripheral blood using the RosetteSep Human CD4+ T Cell Enrichment Cocktail kit (Stemcell Technologies, Vancouver, BC, Canada), following the manufacturer’s instructions, and the purification was efficiency was tested by flow cytometry.

Purified CD4+ T cells were subsequently cultured in 24-well plates, at a density of 1 × 10^6^ cells/mL in RPMI 1640 medium supplemented with 10% FCS (both from Lonza, Basel, Switzerland), and 2 mM L-glutamine at 37 °C in a humidified atmosphere containing 10% CO_2_. Dynabeads^®^ Human T-activator CD3/CD28 (Gibco Laboratories, Paisley, Great Britain) were used to activate cells, at a cell:bead ratio of 1:1, together with recombinant interleukin-2 (IL-2) at 40 U/mL. Cells were cultured under these conditions for 5 days before analysis.

### 4.4. Preparation of Cell Lysates and Western Blotting

Lentivirally transduced Jurkat cells were starved in RPMI 1640 without FCS for 3 h prior to stimulation with anti-CD3 monoclonal antibody (mAb) at 37 °C. Cells were then lysed at 2.0 × 10^7^ cells/mL in 2× Laemmli buffer, followed by incubation at 99 °C for 5 min and sonication. For Western blotting, whole-cell lysates were separated by SDS-PAGE and transferred to PVDF membranes, which were incubated with the indicated primary antibodies, followed by the appropriate secondary antibody conjugated to IRDye 800 CW (Li-Cor, Lincoln, NE, USA) or horseradish peroxidase (HRP). Reactive proteins were visualized using the Odyssey CLx Infrared Imaging System (Li-Cor, Lincoln, Nebraska USA), or by enhanced chemiluminescence (ECL) acquired in a ChemiDoc Touch Imaging System (Bio-Rad Laboratories, Hercules, CA, USA). For reprobing, PVDF membranes were incubated for 10 min at room temperature with WB Stripping Solution (Nacalai Tesque, Kyoto, Japan), followed by a TTBS wash.

### 4.5. Ca^2+^ Mobilization Assays

Measurement of intracellular free Ca^2+^ was carried out using Indo-1 AM (acetoxyme-thyl) (2 μM; Molecular Probes, Invitrogen) as previously described [31]. Calcium measurements were performed using a Synergy MX Multi-Mode Reader (Biotek, Winooski, VT, USA) at 37 °C. Cells were excited by light at a wavelength of 340 nm, and the fluorescence emitted at 405 and 485 nm was collected alternately per second. Calcium mobilization was evaluated by the ratio of 405/485 nm fluorescence signal.

### 4.6. Statistical Analysis

Statistics were processed with Microsoft Excel using a two-tailed *t*-test. Levels of significance *p* < 0.05 are presented as *, and *p* < 0.01 as **.

## Figures and Tables

**Figure 1 ijms-24-04574-f001:**
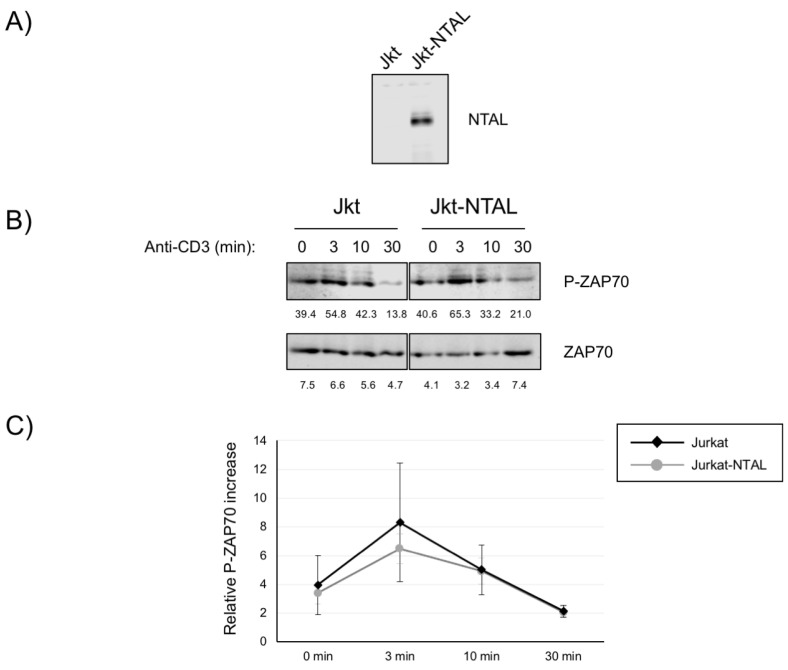
The effect of NTAL expression on ZAP70 phosphorylation. (**A**) NTAL expression in Jurkat cells (Jkt) or Jurkat cells lentivirally transduced with a vector coding for human NTAL (Jkt-NTAL) was assessed by Western blot of cell lysates. A representative experiment out of a total of five is shown. (**B**) Phosphorylation of ZAP70 at tyrosine residue 319 in Jurkat or Jurkat cells expressing NTAL stimulated at 37 °C with soluble anti-CD3 was detected by Western blot using antibodies against pY319-ZAP70 (upper panel) and total ZAP70 (lower panel). One representative experiment is shown. The numbers below each image represent the densitometric quantification of individual bands. (**C**) Diagram representing the quantification of four independent experiments using Jurkat cells (black) or Jurkat cells expressing NTAL (gray line). Phosphorylation levels were normalized to total ZAP70 expression and the means of fold increase of phosphorylation with regard to non-stimulated cells were graphed. Bars represent the standard error.

**Figure 2 ijms-24-04574-f002:**
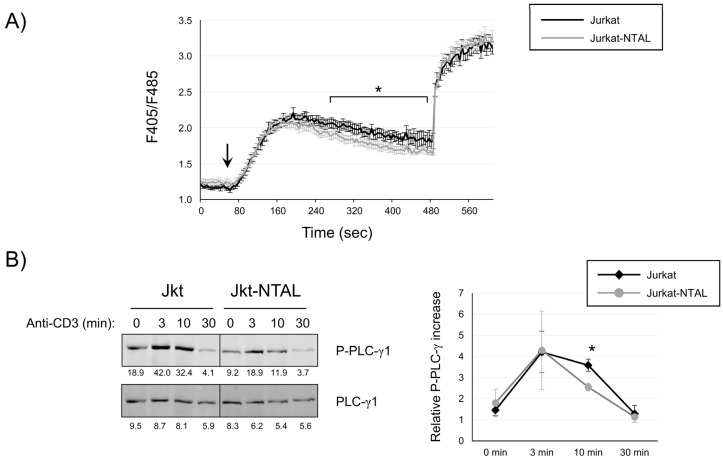
NTAL expression has a negative effect on calcium influx generation and PLC-γ1 phosphorylation. (**A**) Untransduced Jurkat cells or Jurkat cells expressing NTAL were loaded with Indo-1AM and stimulated with OKT3 mAb (1 μg/mL) at the indicated time (arrow). The intracellular Ca^2+^ concentration was determined at 37 °C through the change in Indo-1AM fluorescence. The mean of 12 independent experiments is shown. Bars represent the standard error. Asterisks indicate where statistical significance was less than 0.05. (**B**) Jurkat and Jurkat-NTAL cells were stimulated at 37 °C with anti-CD3 antibody for the indicated times, and whole-cell lysates were probed by Western blotting for the activation of PLC-γ1 by using a mAb-recognizing PLC-γ1 phosphorylated on tyrosine 783 (upper left panel). Stripped membranes were blotted with anti-PLC-γ1 mAb to show equal protein expression (lower left panel). Densitometric quantification of four independent experiments is shown in the diagram on the right, representing mean fold increase in PLC-γ1 phosphorylation for Jurkat (black line) or Jurkat-NTAL cells (gray line). The asterisk represents statistical significance (*p* < 0.05) using a two-tailed *t*-test.

**Figure 3 ijms-24-04574-f003:**
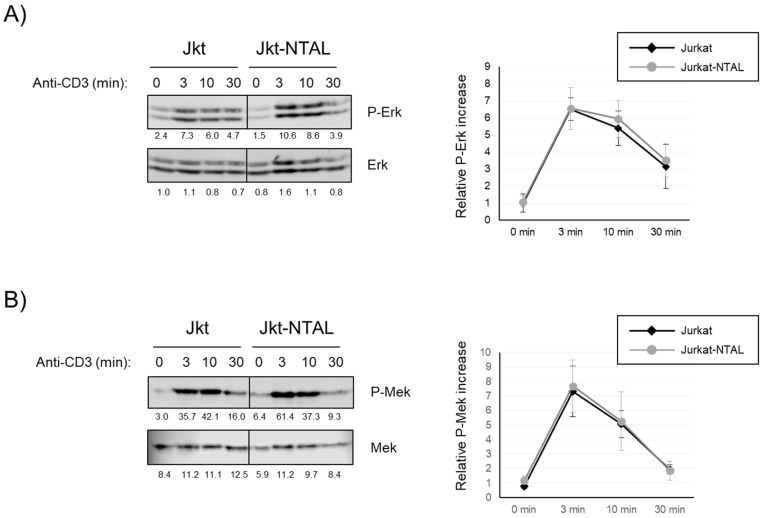
NTAL expression in Jurkat cells does not affect Erk and Mek activation. (**A**) Analysis of phosphorylation of Thr202 and Tyr204 residues in Erk in Jurkat or Jurkat-NTAL cells. Cell lysates were obtained after stimulation at 37 °C with soluble anti-CD3 and Erk phosphorylation was analyzed by Western blot using phospho-specific antibodies (top-left panel) and total Erk (bottom-left panel). One representative experiment is shown. The numbers below each image represent the densitometric quantification of individual bands. The diagram on the right represents the quantification of four independent experiments using Jurkat cells (black) or Jurkat cells expressing NTAL (grey line). Phosphorylation levels were normalized to total Erk expression and the means of fold increase of phosphorylation with regard to non-stimulated cells were graphed. (**B**) Analysis of Mek phosphorylation in CD3-stimulated Jurkat and Jurkat-NTAL cells. Similar to (**A**), cell lysates obtained were analyzed by western blotting with phospho-specific antibodies (bottom-left panel), and the stripped membranes were then blotted with anti-Mek antibodies. The diagram on the right represents the quantification of four independent experiments using Jurkat cells (black) or Jurkat cells expressing NTAL (gray line). The phosphorylation levels were normalized to total Mek expression and the means of fold increase of phosphorylation with regard to non-stimulated cells were graphed. Bars represent the standard error.

**Figure 4 ijms-24-04574-f004:**
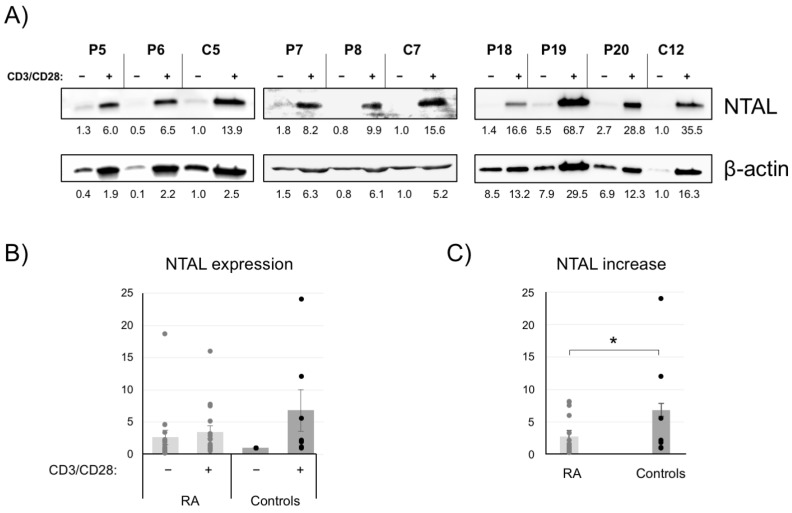
Increased NTAL expression induction in CD4+ T lymphocytes from RA patients. (**A**) NTAL expression analyzed by Western blot in the cell lysates of resting or CD3/CD28-stimulated CD4+ T cells from healthy controls (C) or RA patients (P). Membranes were stripped and reblotted with anti-β-actin antibody to normalize NTAL expression. (**B**) Graphics depicting the relative expression of NTAL in resting or CD3/CD28-activated CD4+ T cells from 16 RA patients and 7 healthy controls. In every single experiment a value of 1.0 was assigned to the corresponding control to determine the relative level of NTAL and β-actin expression. Circles represent individual data points. Bars represent the standard error. (**C**) Graphics representing the relative increase in NTAL expression in CD4+ T cells from RA patients and healthy controls. Bars represent the standard error. The asterisk represents statistical significance (*p* < 0.05) using a two-tailed *t*-test.

**Figure 5 ijms-24-04574-f005:**
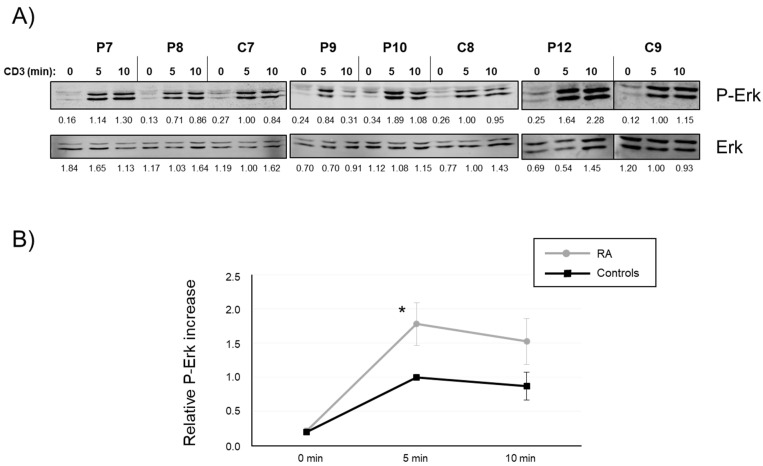
Increased Erk phosphorylation in CD4+ T cells from RA patients. (**A**) Purified CD4+ T cells from RA patients (P) and healthy controls (C) were cultured for 5 days in the presence of anti-CD3/CD28 beads and IL-2, and then stimulated with anti-CD3 mAb at 37 °C for the indicated times. Cell lysates were obtained and phosphorylation of Thr202 and Tyr204 residues in Erk was analyzed by Western blot using phospho-specific antibodies (top panels). Membranes were stripped and total Erk was analyzed to show total protein load (bottom panels). Three independent experiments are shown. Numbers below the panels represent the densitometric quantification of individual bands. (**B**) Graphics representing the quantification of five independent experiments with five healthy controls (black line) and nine RA patients (grey line). Phosphorylation levels were normalized to total Erk expression and the means of fold increase of phosphorylation with regard to non-stimulated cells were graphed. Bars represent the standard error. The asterisk represents statistical significance (*p* < 0.05) using a two-tailed *t*-test.

## Data Availability

Not applicable.

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
