# Peer review of "Expression of Non-T Cell Activation Linker (NTAL) in Jurkat Cells Negatively Regulates TCR Signaling: Potential Role in Rheumatoid Arthritis"

_ijms, 2023, doi:10.3390/ijms24054574_

Round 1
Reviewer 1 Report
In this paper, the authors show that NTAL appears to drive a relatively minor reduction in calcium influx after TCR stimulation. They also show that patients with RA have reduced TCR-induced upregulation of NTAL and reduced TCR-mediated ERK activation. These data add to the current literature suggesting that NTAL is a negative regulator of T cell function. There are several concerns listed below that would strengthen the conclusions of the paper.
1) It is unclear from the Materials and Methods exactly what cells are being used here. The previous study used JCaM2.5 cells which are LAT deficient. Does this study use the same cells or are the authors using wild-type Jurkat E6.1 cells? Are the authors using bulk populations of selected cells or using single cell clones? This information is needed to fully interpret the data.
2) If the cells are distinct from what was described previously, the level of NTAL expression needs to be shown. As of now, it is unclear what levels of NTAL are expressed in these cells.
3) The differences in calcium influx and PLC-gamma1 phosphorylation are relatively minor. This could be the difference between cell clones or in the presence or absence of LAT in the cells. Further information on the cell origin will help clear this up. Additionally, a measure of functionality, such as IL-2 production or changes in protein expression of a marker like CD69 or Nur77, would be useful to determine if the relatively small changes have an impact on effector functions. The addition of functional data is needed to support the authors claims that NTAL is a negative regulator of T cell function.
4) It would be useful to show the individual data points for the data shown in Figure 4B and 4C.
Author Response
1) It is unclear from the Materials and Methods exactly what cells are being used here. The previous study used JCaM2.5 cells which are LAT deficient. Does this study use the same cells or are the authors using wild-type Jurkat E6.1 cells? Are the authors using bulk populations of selected cells or using single cell clones? This information is needed to fully interpret the data.
We thank the reviewer for this comment. In the new version of our manuscript it is clearly specified, in the Results and Material and Methods sections, that we used Jurkat cells for this work. They were not clones, but lentivirally transduced cells, which implies that we had polyclonal populations expressing NTAL or not.
2) If the cells are distinct from what was described previously, the level of NTAL expression needs to be shown. As of now, it is unclear what levels of NTAL are expressed in these cells.
In the new version of our manuscript, we clarify that Jurkat cells do not express the NTAL adaptor. We have included now a Western blot image showing the levels of NTAL expression in Jurkat and lentivirally transduced Jurkat cells (Fig. 1A).
3) The differences in calcium influx and PLC-gamma1 phosphorylation are relatively minor. This could be the difference between cell clones or in the presence or absence of LAT in the cells. Further information on the cell origin will help clear this up. Additionally, a measure of functionality, such as IL-2 production or changes in protein expression of a marker like CD69 or Nur77, would be useful to determine if the relatively small changes have an impact on effector functions. The addition of functional data is needed to support the authors claims that NTAL is a negative regulator of T cell function.
We agree with the reviewer that the differences in PLC-γ activation and calcium fluxes are not very large. The manuscript has been changed to state that our data suggest, in conjunction with previously published data, that NTAL expression may act as a negative regulator of TCR signals. CD69 expression is difficult to analyze, because CD69 levels are very low in Jurkat cells stimulated with anti-CD3 antibodies. However, we consider that our data show that early intracellular signals (PLC-γ and Calcium influx generation) are negatively affected. Future work in the group will address effects on downstream signals.
4) It would be useful to show the individual data points for the data shown in Figure 4B and 4C.
The new version of our manuscript incorporates individual data points in Figures 4B and 4C.
Reviewer 2 Report
In this paper, the authors explored the expression and regulatory functions of NTAL in Jurkat cell line and primary T cells from RA patients and healthy controls (HCs). Here are my questions/ comments.
1. Abstract: The abstract should describe the study objectives, methods, results and conclusions in a concise way. The authors did not clarify these parts.
2. Introduction: there was a typo regarding IFN- γ at line 70. Please correct.
3. Methods:
a. How was the purity of CD4+ T cells after enrichment? What was the percentage of purified CD4+ T cells?
b. Please clarify what type of t- test the authors used. One-sample, two-sample, or paired t test?
4. Results:
a. Now that there were no difference regarding ZAP70 expression between Jurkat and Jurkat-NTAL cells, what did the author believe were the upstream regulators of NTAL?
b. How did the author explain the result that there was difference regarding p-PLC-γ1 levels between Jurkat and Jurkat-NTAL cells at 10min but not at 30min?
c. The authors reported the NTAL expression in activated primary CD4+ T cells from both RA patients and HCs. The Jurkat cell line is CD8- cell line. How was the NTAL expression in CD8+ T cells?
d. What were the characteristics of RA patients and healthy controls, specifically for the RA patients? What were the demographic data? Clinical data? Disease activities and treatment?
e. Was the NTAL expression correlated with disease activities in RA patients?
f. Was there association between NTAL expression and Erk activation in RA patients? Did the authors assess the activation markers like CD69, CD25 and HLA-DR expression at resting status of the T cells in these RA patients?
g. Why did not the authors assess the activation PLC-γ1 as did for Erk?
5. Discussion:
a. There was much background information regarding LAT and NTAL, which was already described in the introduction. Please make it concise.
b. The authors did not summarize the limitations of this study.
Author Response
1) Abstract: The abstract should describe the study objectives, methods, results and conclusions in a concise way. The authors did not clarify these parts.
We are grateful to reviewer 2 for this comment. Consequently, we have modified the abstract according to her/his suggestions.
2) Introduction: there was a typo regarding IFN- γ at line 70. Please correct.
We thank reviewer 2 for the comment. The typo has been corrected.
3) Methods:
a. How was the purity of CD4+ T cells after enrichment? What was the percentage of purified CD4+ T cells?
In the new version of the manuscript we have included the mean percentage of CD3+CD4+ cells was higher than 85%. This has been included in the results section.
b. Please clarify what type of t- test the authors used. One-sample, two-sample, or paired t test?
We apologize for not including this information in the figure legends. In the new version of the manuscript the type of t-test is now included.
4) Results:
a. Now that there were no difference regarding ZAP70 expression between Jurkat and Jurkat-NTAL cells, what did the author believe were the upstream regulators of NTAL?
ZAP70 is the kinase that phosphorylates NTAL and LAT adaptors (since Syk is not expressed in Jurkat cells). Therefore, the negative role that NTAL expression exerts in T cells may be due to the competition with LAT for localization in lipid rafts, as previously suggested by others (Zhu et al., 2006, Immunity: 25: 757-68). The new version of the manuscript discusses this relevant question (lines 354-361).
b. How did the author explain the result that there was difference regarding p-PLC-γ1 levels between Jurkat and Jurkat-NTAL cells at 10min but not at 30min?
Although NTAL is able to activate MAP-kinases pathway after CD3 stimulation, the absence of a PLC-γ binding motif could be the reason for the decreased PLC-γ1 activation and Ca2+ influx generation. This relevant question (related to the previous one) (lines 354-361).
c. The authors reported the NTAL expression in activated primary CD4+ T cells from both RA patients and HCs. The Jurkat cell line is CD8- cell line. How was the NTAL expression in CD8+ T cells?
We have focused on CD4+ T cells because of their relevance for the coordination of immune responses, and their role in RA. This is an important point that we will address in future works.
d. What were the characteristics of RA patients and healthy controls, specifically for the RA patients? What were the demographic data? Clinical data? Disease activities and treatment?
In the new version of our manuscript we have included these data.
e. Was the NTAL expression correlated with disease activities in RA patients?
Our study has been a first approach to test the potential involvement of NTAL as a regulator of T-cell activation and its possible implication in rheumatoid arthritis. The number of patients studied is too small to establish a correlation with the stage of the disease. Future work should clarify this point.
f. Was there association between NTAL expression and Erk activation in RA patients? Did the authors assess the activation markers like CD69, CD25 and HLA-DR expression at resting status of the T cells in these RA patients?
The Erk activation experiments were performed on a small group of patients, from which a sufficient number of CD4+ T cells were obtained to perform the analysis of Erk activation. Unfortunately, the sample size does not allow this type of correlation to be established.
g. Why did not the authors assess the activation PLC-γ1 as did for Erk?
PLC-γ1 activation is much more difficult to analyze in primary cells than Erk, and we decided to study Erk activation given the ease of its analysis by Western blot with a relatively small number of cells.
5) Discussion:
a. There was much background information regarding LAT and NTAL, which was already described in the introduction. Please make it concise.
Thank you very much for pointing this out. We have reduced the text in the discussion section so as not to repeat concepts already explained in the introduction.
b. The authors did not summarize the limitations of this study.
We have included in the Discussion section two limitations of our study, which are the small number of patients, and the lack of study of NTAL expression on CD8+ T cells. Future work should address these.
Round 2
Reviewer 1 Report
The authors have sufficiently addressed my comment and concerns.
Reviewer 2 Report
The authors have addressed my concerns properly. I have no further questions.